# Co-Stimulatory and Immune Checkpoint Molecule Expression on Peripheral Immune Cells Differs Age Dependently Between Healthy Donors and Patients with Head and Neck Squamous Cell Carcinoma

**DOI:** 10.3390/cancers17193215

**Published:** 2025-10-02

**Authors:** Lisa K. Puntigam, Ayla Grages, Julius M. Vahl, Franziska Oliveri, Matthias Brand, Simon Laban, Jens Greve, Marie N. Theodoraki, Cornelia Brunner, Thomas K. Hoffmann, Patrick J. Schuler, Adrian Von Witzleben

**Affiliations:** 1Department of Otorhinolaryngology, Head & Neck Surgery, Ulm University Medical Center, Frauensteige 12, 89075 Ulm, Germany; 2Core Facility Immune Monitoring, Ulm University, 89075 Ulm, Germany

**Keywords:** immune checkpoints, HNSCC, immune senescence, ageing, targeted immunotherapy

## Abstract

In this study, we examined how aging affects the immune system in individuals with and without head and neck cancer. In healthy individuals, certain immune markers declined with age, whereas in cancer patients these markers increased. The most pronounced changes involved molecules that regulate immune activation and inhibition. Our findings suggest that age is an important factor to consider when designing immunotherapy strategies for cancer patients.

## 1. Introduction

Head and neck squamous cell carcinoma (HNSCC) affects more than 2.5 million people and results in 379,000 deaths per year, making it the sixth most common type of cancer worldwide [1]. HNSCC accounts for more than 90% of all malignant tumors in the head and neck region. This entity includes oropharyngeal squamous cell carcinoma (OPSCC) as well as squamous cell carcinomas (SCCs) of the oral cavity, larynx, and hypopharynx. The main risk factors are tobacco and alcohol use, while in OPSCC, infection with human papillomavirus (HPV) represents an additional causative factor [2,3,4].

Inhibitory antibodies targeting immune checkpoint molecules (ICMs), particularly PD-1, have become an established standard of care in recurrent or metastatic (R/M) HNSCC. They are applied either as monotherapy or, in platinum-naïve patients, in combination with cisplatin and 5-fluorouracil, while in platinum-refractory disease monotherapy is the preferred option [5,6,7,8].

However, the immune checkpoint expression on peripheral blood cells is also an important factor for understanding the regulation of the patient’s immune system and their response to an antibody cancer treatment. When these checkpoints are dysregulated systemically, it can lead to inappropriate immune responses, which can contribute to the development of autoimmune diseases, allergies, and most importantly cancer.

The assessment of the tumor-infiltrating lymphocytes (TIL) status in solid tumors has gained more attraction and can be evaluated inter alia by CD8, CD4, or CD3 immunohistochemical (IHC) staining. Additionally, examining the expression of exhaustion markers in TIL is of great interest regarding their activation status and response to therapeutic antibodies in cancer treatment. Well-characterized co-stimulatory molecules include CD137, OX40, glucocorticoid-induced TNF receptor family-related protein (GITR), and CD27, whereas immune checkpoint molecules (ICM) comprise programmed death-1 (PD-1), cytotoxic T-lymphocyte–associated protein 4 (CTLA-4), B- and T-lymphocyte attenuator (BTLA), lymphocyte activation gene 3 (LAG-3), and T-cell immunoglobulin and mucin domain-containing protein 3 (TIM-3).

The immune status of an individual person is dependent on their age. The immune system undergoes significant changes throughout life, which can affect the ability to respond to infections and tumor cells [9]. There exists a controversial phenomenon called immunosenescence, characterized by immunological decline, attributed to a multitude of factors encompassing alterations in the thymus gland, a vital organ responsible for T cell maturation, diminished generation of novel immune cells, and modifications in the functionality of pre-existing immune cells. This can lead to increased susceptibility to infections and a decreased ability to respond to vaccination. Therefore, with age humans have an increased risk of developing certain diseases, such as cancer and autoimmune disorders, which is relevant as the mean age of HNSCC patients is increasing [10].

Summarized, there is a need to reveal the immune checkpoint molecule expression on peripheral blood cells as an important factor in the regulation of the immune system, which can give superior information about the patient’s immune status, especially if compared between tumor and healthy patients. We aimed to investigate how ICM expression differs between HNSCC patients and healthy donors depending on age and to identify potential prognostic ICM markers for cancer immunotherapy in the elderly.

## 2. Material and Methods

### 2.1. Case Selection

This retrospective study was approved by the ethics committee of the University of Ulm (#90/15). Peripheral blood samples were collected from 30 healthy donors and 37 patients with HNSCC treated at the local ENT department (Table 1). All tumor patients were treatment-naïve at the time of sampling and subsequently received therapy according to their respective tumor stage. The HPV status was analyzed in routine diagnostics through pathology. The inclusion criteria for tumor patients included histopathological diagnosed HNSCC, no prior cancer diagnosis, and no prior cancer treatment. Healthy patients were characterized as patients without cancer diagnosis prior to a non-cancer related surgery in the ENT department (detailed characteristics in Appendix A). All patients signed the informed consent form approved by the local ethical review committee.

### 2.2. Isolation of Peripheral Blood Mononuclear Cells (PBMC)

Blood samples (40–50 mL) from both cancer patients and healthy donors were collected in citrate anticoagulation tubes. Peripheral blood mononuclear cells (PBMCs) were isolated using the Leucosep System (Greiner Bio-One, Frickenhausen, Germany) following the manufacturer’s instructions. For preparation, Leucosep tubes were filled with 15 mL Biocoll separation medium (Biochrom, Berlin, Germany) and centrifuged at 1000× *g* for 1 min. Whole blood samples were then layered directly onto the prepared tubes and centrifuged at 800× *g* for 15 min. After centrifugation, the PBMC-containing interphase was carefully collected. The cell fraction was washed once with phosphate-buffered saline (PBS) by centrifugation at 400× *g* for 10 min. Following removal of the supernatant, PBMCs were resuspended in 10 mL PBS and quantified using an automated cell counter (Bio-Rad, Hercules, CA, USA).

### 2.3. Flow Cytometry

Up to 1 × 10^7^ cells were incubated with monoclonal antibodies (mAbs) at room temperature for 30 min in the dark, then washed and resuspended in PBS containing 5% BSA for flow cytometry analysis. Flow cytometry was carried out using a Gallios flow cytometer with Kaluza software version 2.1 (Beckman Coulter, Brea, CA, USA).

The following anti-human mAbs were used for staining: CD8-FITC, CD137-PE, CD39-PE-Cy7, CD4-AF700, LAG3-PE, PD1-PE (all eBioscience, Waltham, MA, USA), BLTA-BV421, GITR-BV421, OX40-PE-Cy5, TIM3-PB (all BioLegend, London, UK), and CD45-AMCyan (BD Biosciences, San Jose, CA, USA). Detailed staining panels and antibody information are provided in Appendix A. All antibodies were pre-titrated on both stimulated and unstimulated peripheral blood lymphocytes to determine optimal staining concentrations.

### 2.4. Gating Strategy

The lymphocyte population was gated based on characteristic forward and side scatter properties. Cytotoxic T cells were identified as CD8+ cells, as previously described by our group [11]. T helper cells were defined as CD4+ cells, and regulatory T cells (Tregs) were defined as CD4 + CD39+ cells, following previous studies that characterized this subset as highly immunosuppressive [12]. The complete gating strategy is shown in Appendix A.

### 2.5. Analyses and Statistics

The data were analyzed using Microsoft Office Excel for Mac (vers. 16.45; Redmond, Washington, DC, USA) and GraphPad Prism (vers. 9; San Diego, CA, USA). Statistical analyses were performed using correlation analyses with Pearson’s r value. In the analyses *p*-values with lower than 0.05 are considered statistically significant.

## 3. Results

### 3.1. Patient Characteristics

The patient cohort used for analysis consisted of 30 healthy controls (female: n = 18; male: n = 12) and 37 HNSCC patients (female: n = 12; male: n = 25). The median age of the control group was 49 years (range: 21–84 years); for the HNSCC patients it was higher at 62 years (range 37–94 years). Of the included 11 oropharyngeal cases, 7 were classified as HPV^pos^. The remaining oropharyngeal cases and the tumors from the other locations were classified as HPV^neg^ (n = 20). The detailed characteristics are provided in Appendix A. A summary of the clinicopathological features of the healthy donors is provided in Appendix A.

### 3.2. ICM and Co-Stimulatory Molecule Expression on CD8^+^ T Cells Differs Between Healthy Donors and HNSCC Patients Depending on Their Age

In healthy donor PBMCs, ICM expression on CD8^+^ T cells shows a statistically significant decrease with increasing age for LAG3, PD1, and BTLA (LAG3: *p* = 0.0282; PD1: *p* = 0.0072; BTLA: *p* = 0.0027) while for TIM3 and CTLA4 no statistically significant changes were detected. In PBMCs from HNSCC patients, there is a tendency of increasing ICM expression depending on their age (Figure 1A). For the co-stimulatory molecules, we could find a statistically significant decrease in CD137 (*p* = 0.0470) on the healthy PBMCs while there was no difference in the expression of the HNSCC immune cells. Interestingly, CD27 was the only co-stimulatory molecule that was equally reduced expressed on PMBCs of healthy as well as HNSCC patients (healthy: *p* < 0.0001; HNSCC: *p* = 0.0168; Figure 1B; non-statistically significant pairs are provided in the Appendix A).

### 3.3. ICM and Co-Stimulatory Molecule Expression on CD4^+^ T Cells Varies Between Healthy Donors and HNSCC Patients Depending on Their Age

Consistent with the CD8^+^ T cell data, the ICM expression on CD4^+^ T cells has a statistically significant decrease with increasing age in the healthy donor PBMCs for LAG3, BTLA, and—different to CD8^+^—CTLA4 (LAG3: *p* = 0.0349; BTLA: *p* = 0.0230; CTLA4: *p* = 0.0470), while for PD1 and TIM3 no statistically significant changes were detected. In contrast, the expression of the ICM PD1, LAG3, and CTLA4 (PD1: *p* = 0.0005; LAG3: *p* = 0.0321; CTLA4: *p* = ns) revealed a positive correlation with increasing age in HNSCC patients (Figure 2A). Intriguingly, the BTLA expression showed, similar to the healthy donors, a significant decrease in expression with increasing age (*p* = 0.0019; Figure 2A).

The co-stimulatory molecule expression of GITR (*p* = 0.0404) on CD4^+^ T cells was statistically significantly decreased on the healthy PBMCs, while there was no difference in the expression of the HNSCC immune cells. Corresponding to the CD8^+^ T cell data, CD27 was again the single co-stimulatory molecule that was equally reduced, expressed on PBMCs of healthy as well as on PMBCs of HNSCC patients (healthy: *p* = 0.0219; HNSCC: *p* = 0.0018; Figure 2B; non-statistically significant pairs are provided in the Appendix A).

### 3.4. Immunosuppressive CD39^+^ T Cells Exhibit a Different ICM and Co-Stimulatory MOLECULE Expression Pattern

CD39^+^ T cells in HNSCC patients as well as healthy donors have a reduction in BTLA expression with increasing age. Nevertheless, this tendency was proven to be significant only for the healthy donors (*p* = 0.001; Figure 3A). CTLA4 expression revealed a reverse correlation between expression and age (healthy CTLA4: *p* = 0.0475; Figure 3A). No statistically significant changes are seen for PD1—for comparison it is presented in Figure 3. The co-stimulatory molecule expression on CD39^+^ T cells revealed a statistically significant decrease in GITR and CD137 (*p* = 0.0209, *p* = 0.0183, respectively; Figure 3B) on healthy PBMCs while there was a significant decrease on HNSCC PMBCs for CD27 expression with increasing age (*p* = 0.0205). The expression of CD27 in CD39^+^ T cells increases in both healthy individuals and tumor patients, whereas the CD8^+^ and CD4^+^ T-cell populations exhibit an opposite trend between the two groups (Figure 3B; non-statistically significant pairs are provided in the Appendix A).

We provided a result summary of all *p*-values and correlation coefficients in a table format for each analysis in Appendix A.

### 3.5. HPV Status Has a Significant Influence on ICM and Co-Stimulatory Molecule Expression

The cohort of HNSCC cases was dichotomized in HPV^pos^ OPSCC patients and HPV^neg^ HNSCC patients (HPV^neg^ OPSCC and all other locations). We were able to demonstrate a statistically significant decrease in BTLA in HPV^neg^ PBMC (*p* = 0.0009) while there was no change in HPV^pos^ PBMC for ICM on CD4^+^ T cells. Interestingly, CTLA4 as an ICM was significantly increased on CD8^+^, CD4^+^, and CD39^+^ T cells of the HPV^pos^ but not for the HPV^neg^ group (Figure 4A). Both HPV^pos^ and HPV^neg^ HNSCC PBMC showed a significant increase in PD1 expression (*p* = 0.0206, *p* = 0.0052; Figure 4A). OX40 expression was also significantly increased with age in HPV^pos^ cases, while HPV^neg^ had no change in expression.

There was an equal expression of the co-stimulatory molecule CD27 on CD8^+^, CD4^+^, and CD39^+^ T cells for HPV^pos^ OPSCC. However, the expression was decreasing with higher age in HPV^neg^ HNSCC PBMC (*p* = 0.0494, *p* = 0.0070, and *p* = 0.0258, respectively; Figure 4B). No differences between the HPV status were seen for CD137, GITR, LAG3, or TIM3.

We provide a summary of all expression analysis results including the respective statistics in Appendix A.

### 3.6. ICM and Co-Stimulatory Molecule Expression Differs Between the Sexes

We conducted an additional sex-specific analysis of both cohorts to evaluate potential differences in ICM and co-stimulatory molecule expression. Regarding the ICM PD-1, LAG-3, and CTLA-4, the increase in expression in the tumor group was more pronounced in male participants (Appendix A). We barely saw expression differences between females and males in the healthy donors, whereas the expression of co-stimulatory molecules, such as CD27, showed the same trends in both cohorts in the sex-specific analysis as in the overall evaluation (Figure 5). An separate analysis of the CD4/CD8 ratio was consistently 3:1 in both women and men, with no differences observed between the control group and the tumor patients.

## 4. Discussion

In this study, we aimed to investigate the expression of co-stimulatory and immune checkpoint molecules (ICMs) on peripheral blood cells in healthy individuals and cancer patients across different age groups. Our findings highlight significant age-related shifts in immune regulation, particularly in the expression of CTLA-4, CD27, and BTLA, which could influence treatment strategies for older HNSCC patients.

### 4.1. Age-Related Differences in Co-Stimulatory Molecules/ICM Expression and Implications

Aging is associated with a decline in adaptive immunity, facilitating tumor immune evasion [9]. The weakened immune response to cancer in the elderly may also stem from various alterations in immune cell populations. Age-related immune senescence, characterized by diminished T cell function and fewer tumor-infiltrating immune cells, further reduces the effectiveness of immune checkpoint blockade in HNSCC [13]. Additionally, the expression patterns of immune checkpoint molecules and co-stimulatory signals impair T cell activation, weakening the anti-tumor response. Age also influences the tolerability and toxicity of immune checkpoint inhibitors, as older patients—who often have more comorbidities and a lower physiological reserve—are more prone to treatment-related adverse events, particularly skin-related toxicity, while younger patients are more likely to experience endocrine side effects [14].

In our study, we examined the expression of the ICM CTLA-4, which is constitutively expressed on regulatory T cells (Tregs) and upregulated in conventional T cells upon activation, with its dynamic regulation attracting significant interest in cancer immunology [15]. We found a significant decrease in CTLA-4 expression on CD4^+^ T cells in healthy donors and an increase in cancer patients with age. The upregulation of CTLA-4 on CD4^+^ T cells which may reflect a tendency towards immune suppression and could be relevant for tumor progression [16]. We observed a significantly higher expression of CTLA-4 on CD39^+^ T cells in HPV^pos^ patients with age. These T cells, known for their strong suppressive activity, have an enhanced ability to suppress immune responses due to the increased CTLA-4 expression [12]. The age-associated increase in CTLA-4 expression on CD4^+^ T cells highlight potential implications for cancer immunotherapy and treatment strategies. Immunotherapeutic approaches targeting CTLA-4, such as monoclonal antibodies that block its inhibitory function, have shown promising potential in enhancing anti-tumor immune responses [17,18]. However, the age-related increase in CTLA-4 expression warrants further investigation to assess the potential benefits of such interventions in older individuals.

Our results further highlight the impact of aging on immune function, particularly the loss of co-stimulatory molecules like CD27, which diminishes the capacity to respond to antigens and contributes to resistance to immunotherapy [19]. This reduced CD27 expression with increasing age was observed in both CD8^+^ and CD4^+^ T cells of HNSCC and healthy patients. Interestingly, in HNSCC patients, CD27 expression on CD39^+^ T cells remained stable compared to healthy donors. The co-expression of CD27 and CD39 on Tregs, which has been linked to a highly suppressive phenotype, suggests that this combined expression may enhance the regulatory functions of Tregs, presenting an additional immunosuppressive mechanism in aged HNSCC patients.

In our study, we also investigated the role of BTLA, an immune checkpoint molecule primarily expressed by B and T cells that inhibits activation and signal transduction. BTLA shares similarities with other ICMs like PD1 and CTLA-4, which are current targets of immunotherapies. Our results show that in HNSCC patients, the expression of BTLA on CD8^+^ T cells remains stable at around 60%, whereas in healthy individuals, BTLA expression on PBMCs decreases significantly with age. This might suggests that older HNSCC patients have a more exhausted and dysfunctional cytotoxic T cell population, as published before [20], highlighting potential therapeutic opportunities. Notably, ongoing clinical trials are exploring the effects of anti-BTLA antibodies, both as monotherapy and in combination with anti-PD1 antibodies, for advanced solid malignancies (NCT04137900).

Importantly, these findings might also be transferable to other solid tumors, such as lung cancer or melanoma, where published data already suggest that the efficacy of immune checkpoint inhibitors may be reduced in older patients [21,22,23].

### 4.2. HPV Status and Immune Landscape Differences

Regarding HPV-status and ICM expression, we found that HPV^pos^ HNSCC patients exhibited a more stable or increasing expression of co-stimulatory molecules with age, whereas HPV^neg^ patients showed a decrease. Additionally, HPV^neg^ HNSCC patients had higher expression of co-inhibitory ICMs, such as PD1, which may contribute to immune suppression, which was shown by our group in a sequencing analysis [24]. A positive HPV-status is known to significantly influence tumor immune cell infiltration, enhancing the immune response due to viral antigens, which correlates with improved survival in this group [25,26,27]. These immunological differences explain the significantly better prognosis of HPV^pos^ patients compared to HPV^neg^ patients, as HPV^pos^ tumors tend to have a more favorable immune profile and therefore a better response to immune-based therapies. The increasing incidence of oropharyngeal cancer also among elderly patients, largely attributed to HPV-induced cancers [28,29], highlights the need for further investigation into the age-specific immune landscape in HPV^pos^ HNSCC patients, an area where data are currently lacking.

### 4.3. Sex Differences in Immune Aging

The sex-specific analysis revealed notable differences in ICM expression, particularly PD-1, within the tumor group, with higher expression levels observed in males. It is known that immunological parameters such as the percentage of CD4^+^ naïve T cells, memory CD4^+^ T cells, and NK cells, decline more gradually in women than in men, along with a more favorable CD4^+^/CD8^+^ ratio in women [30]. This pattern seems to extend to ICM expression, potentially leading to a weaker immune response in male HNSCC patients. Since there were fewer women than men in our tumor cohort, this can only be considered a trend. A matched analysis with an equal proportion of men and women would be necessary to confirm this with certainty.

### 4.4. Limitations

As further limitations of this study, we note the sample size of both groups. While we acknowledge that a larger sample could provide more robust data, we believe that our sample size of 30 healthy individuals and 37 cancer patients is adequate for an initial study aimed at identifying key trends and drawing meaningful preliminary conclusions. A potential limitation is that three healthy individuals had chronic inflammatory conditions. However, since these conditions were primarily localized, they are unlikely to significantly affect systemic inflammation or our comparisons. Another important limitation we need to mention is the gating strategy which focused mainly on identifying cytotoxic and immunosuppressive T cells. We did not include other subpopulations like NK-T cells, which might influence our findings. The ICM and co-stimulatory molecule expression in these subpopulations could be an interesting areal for future research.

## 5. Conclusions

In summary, our data show age-related changes in ICM expression across the lymphocyte populations we investigated in tumor patients compared to healthy donors, pointing to alterations in immune regulation that could influence the tumor microenvironment. These observations may provide a rationale for considering immune checkpoint inhibitors in older HNSCC patients, although we emphasize that our findings do not warrant immediate changes to current treatment. Importantly, immune checkpoint inhibitors are not specifically approved for age-related indications, and current guidelines do not offer age-specific recommendations. In our study, we identified ICMs such as CTLA-4, TIM3, and BTLA as promising targets for further investigation, particularly in elderly patients, and we highlight the need for additional research to better understand their predictive and therapeutic potential in HNSCC.

## Figures and Tables

**Figure 1 cancers-17-03215-f001:**
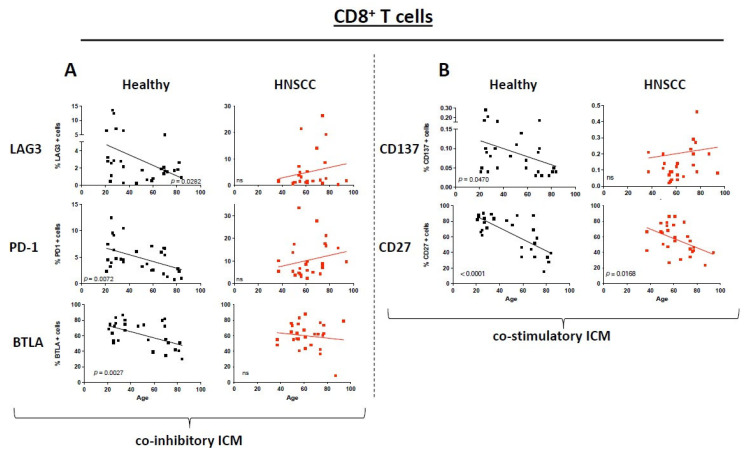
Dot plots depict the expression of immune checkpoint molecules and co-stimulatory molecules on CD8^+^ T cells for healthy donors (black) and HNSCC samples (red). The straight line represents Pearson’s correlation analysis along with the corresponding *p*-values. (**A**): Selected immune checkpoint molecules; (**B**): selected co-stimulatory molecules. *ns* = *not significant*.

**Figure 2 cancers-17-03215-f002:**
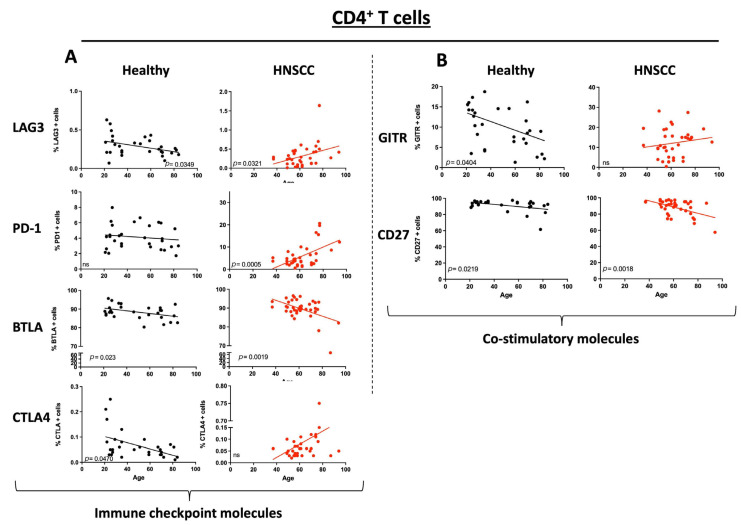
Dot plots depict the expression of immune checkpoint molecules and co-stimulatory molecules on CD4^+^ T cells for healthy donors (black) and HNSCC samples (red). The straight line represents Pearson’s correlation analysis along with the corresponding *p*-values. (**A**): Selected immune checkpoint molecules; (**B**): selected co-stimulatory molecules. *ns* = *not significant*.

**Figure 3 cancers-17-03215-f003:**
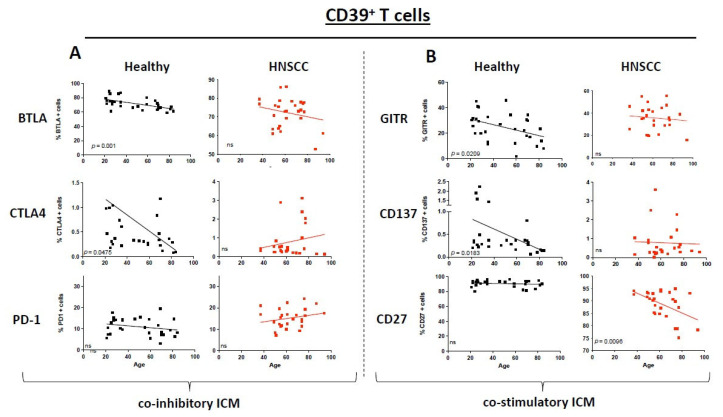
Dot plots depict the expression of immune checkpoint molecules and co-stimulatory molecules on CD39^+^ T cells for healthy donors (black) and HNSCC samples (red). The straight line represents Pearson’s correlation analysis along with the corresponding *p*-values. (**A**): Selected immune checkpoint molecules; (**B**): selected co-stimulatory molecules. *ns* = *not significant*.

**Figure 4 cancers-17-03215-f004:**
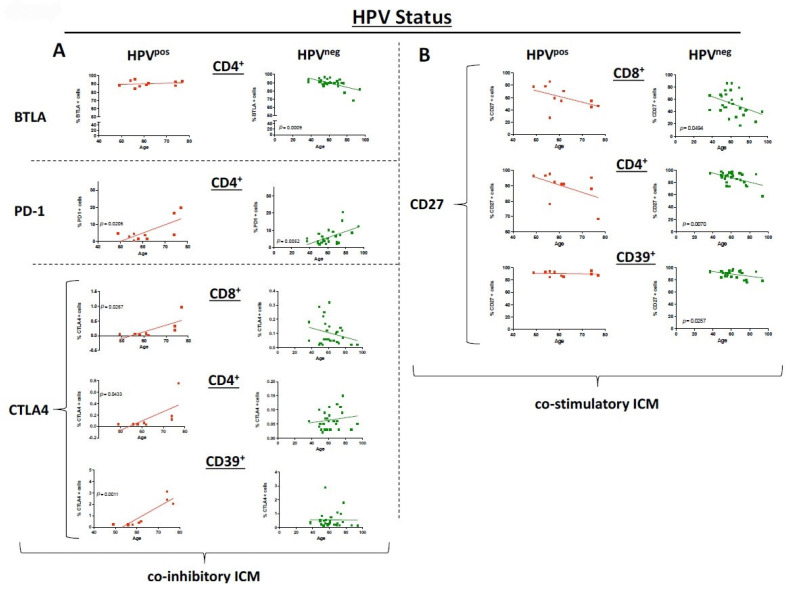
Dot plots depict the expression of selected immune checkpoint molecules and co-stimulatory molecules on different T cells for HNSCC samples grouped by HPV status: HPV^pos^: red, HPV^neg^: green. The straight line represents Pearson’s correlation analysis along with the corresponding *p*-values. (**A**): Selected immune checkpoint molecules; (**B**): selected co-stimulatory molecules. *ns* = *not significant*.

**Figure 5 cancers-17-03215-f005:**
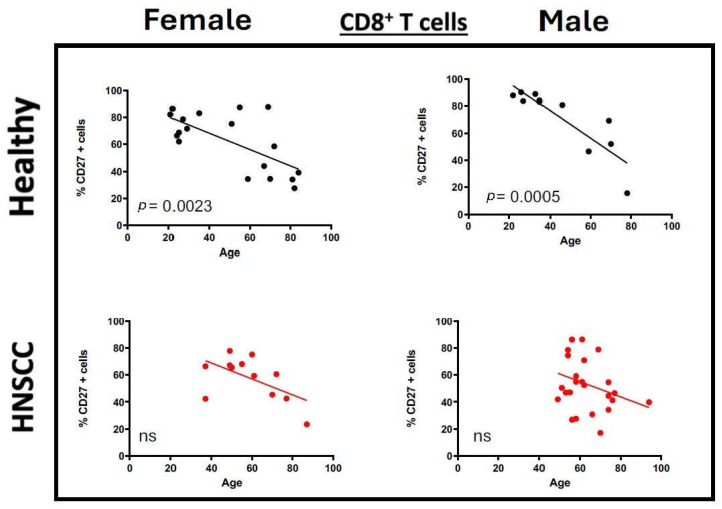
Dot plots depict the expression of the costimulatory molecule CD27 on CD8+ T cells for healthy donors (black) and HNSCC samples (red). The straight line represents Pearson’s correlation analysis along with the corresponding *p*-values. Female and male patients are compared. *ns* = *not significant*.

**Table 1 cancers-17-03215-t001:** Clinicopathological characteristics of HNSCC patients and the control group. Summary of the patient’s characteristics. *HPV* = *human papillomavirus*, *y* = *years*, *HNSCC* = *head and neck squamous cell carcinoma*, *OPSCC* = *oropharyngeal squamous cell carcinoma*.

	Control Group	HNSCC Patients
*N* (female/male)	30 (18/12)	37 (12/25)
Age (±SD) range (y)	49 ± 22 (21–84)	62 ± 12 (37–94)
Stage		n=
T1		8
T2		14
T3		7
T4		8
Nodal status		n=
N0		10
N1		12
N2		6
N3		9
Location		n=
Oral cavity		11
Oropharynx		15
Hypopharynx		5
Larynx		6
HPV status (of OPSCC cases)		n=
Positive		10
Negative		5

## Data Availability

The data are available upon request from the authors.

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
