# Peer review of "Co-Stimulatory and Immune Checkpoint Molecule Expression on Peripheral Immune Cells Differs Age Dependently Between Healthy Donors and Patients with Head and Neck Squamous Cell Carcinoma"

_cancers, 2025, doi:10.3390/cancers17193215_

Round 1
Reviewer 1 Report
Comments and Suggestions for Authors
This is an interesting observational study. The authors collected peripheral blood samples from 30 healthy controls and 37 patients diagnosed with HNSCC, and quantified the expression of co-stimulatory and immune checkpoint molecules by flow cytometry. They further analyzed the correlation between these expression levels and age. Apart from the HPV-stratified subgroup, I did not observe strong trends in the other analyses. Below are my comments and suggestions:
- Underlying comorbidities such as type II diabetes or autoimmune diseases can profoundly affect systemic immune status. Therefore, baseline comorbidity information of the patient cohort should be reported, and hazard ratio (HR) analyses could be performed if possible.
- The authors should check whether the age and sex distributions are comparable between the healthy control and patient groups.
- Given that HPV is a relatively common virus, HPV status should also be assessed in the healthy control group, at least by PCR-based testing.
- I noted that among the healthy controls, only 11 had no prior medical conditions. While some conditions (e.g., tonsillectomy, deviated nasal septum, vocal cord polyp) can reasonably be considered compatible with a “healthy” state, others (e.g., obstructive sleep apnea) may significantly impact immune status. In addition, the three cases with “unknown” past medical history should not be included in the healthy control cohort.
- Both in the healthy control and HNSCC groups, age appears to have some influence on the expression of co-stimulatory and immune checkpoint molecules, though the correlations are not strong. Expanding the sample size in both cohorts may help reveal more robust trends. Of course, only truly healthy samples should be included in the control group, and those with potential comorbidities and unknown status should be excluded.
- The authors state several times that changes in co-stimulatory and immune checkpoint molecule expression directly contribute to tumor development. It may be more appropriate to soften these statements, as immune status and tumor progression are dynamic processes that influence one another. In my opinion, the true value of this study lies in highlighting the potential of these molecular profiles as predictors of immunotherapy response or even prognosis. The discussion could be refined to better emphasize this perspective.
- A minor point: when preparing figures, capturing screenshots in display mode would avoid the appearance of red wavy lines (as seen in Figure 5 and Supplementary Figure 3).
Author Response
Reviewer’s comment 1:
Underlying comorbidities such as type II diabetes or autoimmune diseases can profoundly affect systemic immune status. Therefore, baseline comorbidity information of the patient cohort should be reported, and hazard ratio (HR) analyses could be performed if possible.
Authors’ reply:
Thank you very much for this helpful comment. We agree that comorbidities such as type II diabetes or autoimmune diseases can strongly influence immune status. This is why we selected healthy controls without relevant comorbidities. For the HNSCC patients, comorbidity data can be retrieved, but based on the clinical profile of our cohort, we do not expect this to change the results in a major way.
Reviewer’s comment 2:
The authors should check whether the age and sex distributions are comparable between the healthy control and patient groups.
Authors’ reply:
Thank you for this suggestion. We carefully looked at the age distribution between the groups. Although there is a difference in mean age, we decided to keep the full age range, as this allows us to capture a broader spectrum and possibly reveal age-related trends. In addition, healthy controls are usually easier to recruit at younger ages and without major comorbidities (see comment 1), which makes closer age matching difficult. Unfortunately, it was also challenging to achieve a balanced gender distribution between patients and controls in this setting.
Reviewer’s comment 3:
Given that HPV is a relatively common virus, HPV status should also be assessed in the healthy control group, at least by PCR-based testing.
Authors’ reply:
We agree that HPV is a common virus - however, HPV testing (e.g. with PCR) is only meaningful in tissue samples, which were not collected from the healthy control participants. Therefore, assessing HPV status in these controls is not feasible within the context of our study.
Reviewer’s comment 4:
I noted that among the healthy controls, only 11 had no prior medical conditions. While some conditions (e.g., tonsillectomy, deviated nasal septum, vocal cord polyp) can reasonably be considered compatible with a “healthy” state, others (e.g., obstructive sleep apnea) may significantly impact immune status. In addition, the three cases with “unknown” past medical history should not be included in the healthy control cohort.
Authors’ reply:
We thank the reviewer for this comment. We did not expect the reported prior medical conditions to have a major impact on immune status. Many of these conditions are also quite common among the tumor patients, so any effects tend to balance out. Regarding the three cases with “unknown” medical history, we have now checked the records and can assign them to “None”. Overall, we believe the healthy control group remains appropriate for the comparisons in our study.
The healthy subjects who, for instance, presented for tonsillectomy were not operated on during an acute inflammatory episode (i.e., no tonsillar abscess). Moreover, for conditions such as nasal septum deviation (NSD), no studies to date have demonstrated an effect on immune checkpoint molecules. In patients with obstructive sleep apnoea syndrome (OSAS), a chronic systemic inflammatory response cannot be excluded; however, as this applies to only a small proportion of patients, the potential impact on the overall cohort appears negligible. To exclude a substantial influence of these patients on the dataset, we carefully re-examined their individual results, which did not reveal any marked outliers in either direction.
Reviewer’s comment 5:
Both in the healthy control and HNSCC groups, age appears to have some influence on the expression of co-stimulatory and immune checkpoint molecules, though the correlations are not strong. Expanding the sample size in both cohorts may help reveal more robust trends. Of course, only truly healthy samples should be included in the control group, and those with potential comorbidities and unknown status should be excluded.
Authors’ reply:
We agree that the investigation of ICM expression in relation to patient age is a complex topic requiring extensive research. However, for a study aiming to provide initial results and identify key trends, we believe that our sample size of 30 healthy individuals and 37 cancer patients is sufficient to draw preliminary conclusions. Recruiting elderly patients for the control group who present without any comorbidities - or have matched comorbidities - appears to be difficult to achieve.
Reviewer’s comment 6:
The authors state several times that changes in co-stimulatory and immune checkpoint molecule expression directly contribute to tumor development. It may be more appropriate to soften these statements, as immune status and tumor progression are dynamic processes that influence one another. In my opinion, the true value of this study lies in highlighting the potential of these molecular profiles as predictors of immunotherapy response or even prognosis. The discussion could be refined to better emphasize this perspective.
Authors’ reply:
Thank you for this valid point. We have softened the statements throughout the discussion regarding the causal role of ICM and co-stimulatory molecule expression in tumor development, and we have also revised the conclusion to emphasize their potential as predictive and therapeutic markers rather than direct drivers of tumor progression.
Reviewer’s comment 7:
A minor point: when preparing figures, capturing screenshots in display mode would avoid the appearance of red wavy lines (as seen in Figure 5 and Supplementary Figure 3).
Authors’ reply:
We changed the respective graphs – thanks for spotting this!
Reviewer 2 Report
Comments and Suggestions for Authors
In the manuscript “Co-stimulatory and immune checkpoint molecule expression on peripheral immune cells differs age dependently between healthy donors and patients with head and neck squamous cell carcinoma” Puntigam and colleagues analysed peripheral blood (PB) CD4 and CD8 T cells for the expression of positive and negative regulators of T cell activation that may be altered with aging in health and pathologic conditions. It is already known that in elderly the ability to respond to infection /vaccination is less efficient than in young adults. Identified, as immune senescence, this phenomenon caused by multiple factors may also contribute to inefficient response to tumour cells as soon as they arise and thus strongly contribute for the increased frequency of tumour development with aging.
Here authors study the PB T cell compartment of head and neck squamous cell carcinoma (HNSCC) patients at different ages and found that both in CD4 and CD8 expression of PD-1, LAG-3 increases with age and CD27 decreases, as compared with HD suggesting a reduced ability to fight cancer. The fact that authors, not only address age issues but also gender maybe considered a very positive point of this manuscript.
There are several critical points needing clarification:
- Age groups are not matched, HNSCC are older.
- Number of patients stated on line 84 is not coherent with table 1
- Explain the rational to make staining 30min at RT or give a reference for the used methodology.
- consider to make a supplementary table with antibodies details described at lines 112-115 and give rational to shown “checkpoint panels lines 118-120 in the main manuscript and not as supplement. This info is not referred in the manuscript, could be removed.
- Gating strategy is shown as supplementary figure 1 but is not indicated method section, line 121. The figure should also show how doublets have been selected out and indicate frequencies inside gating. Were Tregs excluded from CD4 pool to identify what authors called T helper cells?
- Scheme 1 that I suppose is the supplementary table 1 is not understandable.
- scheme 2 as point 6 should be revised.
- Improve supplementary figure legend 2, images show also CD4 and CD8 cells and not only CD39+. In addition, some panels are repetitions of the main figures, consider to remove them.
- Supplementary figure 3 shows the same panels of figure 5, wouldn’t be the case to show CD4 compartment or as stated in the text other activation markers?
- Line 312 is stated that CD4/CD8 ratio in women is a favourable factor, is this ratio similar in HD and patients in the analysed cohort? Given the importance of the CD4 compartment I would strongly suggest that gender related analysis of CD4 cells to be added.
- Line 322: for correctness, this information on the healthy donors should be part of table 1.
The conclusion is accurate and in the discussion authors describe the study caveats. I would just add a couple of sentences on how the observations seen and conclusions taken, on HNSCC patients, can potentially be translated to other solid tumours.
Author Response
Reviewer’s comment 1:
Age groups are not matched, HNSCC are older.
Authors’ reply:
As mentioned for Reviewer 1: We carefully looked at the age distribution between the groups. Although there is a difference in mean age, we decided to keep the full age range, as this allows us to capture a broader spectrum and possibly reveal age-related trends. In addition, healthy controls are usually easier to recruit at younger ages and without major comorbidities, which makes closer age matching difficult. Unfortunately, it was also challenging to achieve a balanced gender distribution between patients and controls in this setting.
Reviewer’s comment 2:
Number of patients stated on line 84 is not coherent with table 1
Authors’ reply:
We adjusted the number of patients in table 1.
Reviewer’s comment 3:
Explain the rational to make staining 30min at RT or give a reference for the used methodology.
consider to make a supplementary table with antibodies details described at lines 112-115 and give rational to shown “checkpoint panels lines 118-120 in the main manuscript and not as supplement. This info is not referred in the manuscript, could be removed.
Authors’ reply:
There are several other experimental studies using this staining method, for example:
1.) Jeske SS, Schuler PJ, Doescher J, Theodoraki MN, Laban S, Brunner C, Hoffmann TK, Wigand MC. Age-related changes in T lymphocytes of patients with head and neck squamous cell carcinoma. Immun Ageing. 2020 Feb 12;17:3. doi: 10.1186/s12979-020-0174-7. PMID: 32082401; PMCID: PMC7017629.
2.) Hernández-Campo PM, Martín-Ayuso M, Almeida J, López A, Orfao A. Comparative analysis of different flow cytometry-based immunophenotypic methods for the analysis of CD59 and CD55 expression on major peripheral blood cell subsets. Cytometry. 2002 Jun 15;50(3):191-201. doi: 10.1002/cyto.10072. PMID: 12116342.
In response to this suggestion, we have added a supplementary table detailing all antibodies used. Regarding the checkpoint panels shown in lines 118–120, these were included in the main manuscript to provide an overview of the experimental setup; however, we agree that if not directly referenced in the results, this information can be moved to the Supplementary Material to improve clarity.
Reviewer’s comment 4:
Gating strategy is shown as supplementary figure 1 but is not indicated method section, line 121. The figure should also show how doublets have been selected out and indicate frequencies inside gating. Were Tregs excluded from CD4 pool to identify what authors called T helper cells?
Authors’ reply:
In the FACS analysis, doublets were excluded based on their size in the forward and side scatter.
We adjusted the Supplementary Figure 1 and added the frequencies in the gating strategy.
CD39⁺ cells were not excluded from the CD4⁺ pool, as they represent only a small fraction of the CD4⁺ cell population. Significant differences were observed between the CD4⁺ and CD39⁺ populations; therefore, we expect only minimal influence of CD39⁺ cells on the overall CD4⁺ pool.
Reviewer’s comment 5:
Scheme 1 that I suppose is the supplementary table 1 is not understandable.
scheme 2 as point 6 should be revised.
Authors’ reply:
Supplementary Table 1 summarizes the clinical diagnosis of the control group patients (if they had any). The table has been formatted accordingly.
Reviewer’s comment 6:
Improve supplementary figure legend 2, images show also CD4 and CD8 cells and not only CD39+. In addition, some panels are repetitions of the main figures, consider to remove them.
Authors’ reply:
We adjusted the Figure legend 2.
Reviewer’s comment 7:
Supplementary figure 3 shows the same panels of figure 5, wouldn’t be the case to show CD4 compartment or as stated in the text other activation markers?
Authors’ reply:
We really thank the reviewer for detecting this error. We now included the CD27-Expression on T-cells in the main text and moved the graphs with Tregs and CD4+-T cells in the supplement.
Reviewer’s comment 8:
Line 312 is stated that CD4/CD8 ratio in women is a favourable factor, is this ratio similar in HD and patients in the analysed cohort? Given the importance of the CD4 compartment I would strongly suggest that gender related analysis of CD4 cells to be added.
Authors’ reply:
In our study, the CD4/CD8 ratio was consistently 3:1 in both women and men, with no differences observed between the control group and the tumor patients. This additional information was added to the gender paragraph.
Within the control group, we also conducted a sex-specific analysis of CD4⁺ T cells. This revealed a slightly lower number of CD4⁺ cells in women compared to men; however, the result did not reach statistical significance given the sample size.
Reviewer’s comment 9:
Line 322: for correctness, this information on the healthy donors should be part of table 1.
Authors’ reply:
Thank you for this suggestion. The clinical characteristics of the control group are summarized in the Supplementary Table 1, as their inclusion in Table 1 – we thought – would compromise readability.
Reviewer’s comment 10:
The conclusion is accurate and in the discussion authors describe the study caveats. I would just add a couple of sentences on how the observations seen and conclusions taken, on HNSCC patients, can potentially be translated to other solid tumours.
Authors’ reply:
Thank you for this helpful suggestion. We added a sentence at the end of the first discussion paragraph regarding age-related effects of immunotherapy in other solid tumors such as lung cancer and melanoma.
Reviewer 3 Report
Comments and Suggestions for Authors
I have read with great interest article entitled: Co-stimulatory and immune checkpoint molecule expression on peripheral immune cells differs age dependently between healthy donors and patients with head and neck squamous cell carcinoma. In this study authors found that the expression of ICM in HNSCC was significantly increased (PD1 on CD4+ T cells: p = 0.001), while they found a decreased co-stimulatory molecule expression (e.g., CD27 on CD4+ T cells and CD39+ T 31 cells: p = 0.003 and p = 0.009, respectively). Furthermore, immune cells of HPV neg HNSCC have a significant age-dependent decrease of CD27 expression on CD8+, CD4+ and CD39+ T cells. It is a solid article with results that could potentially enrich current knowledge and clinical practice. I suggest few modifications. There are numerous limitations of this type of study and I encourage authors to explain in more details why these limitations could influence final result (i.e. authors focused mainly on identifying cytotoxic and immunosuppressive T cells, but did not include other subpopulations like NK-T cells - this could potentially influence final results). Also, what would be suggested direction for future studies in the light of your findings? Is there any clinical utility of these findings right now? Please cite literature with the similar topic as your study (compare results and emphasize similarities and differences).
Author Response
Reviewer’s comment 1:
I have read with great interest article entitled: Co-stimulatory and immune checkpoint molecule expression on peripheral immune cells differs age dependently between healthy donors and patients with head and neck squamous cell carcinoma. In this study authors found that the expression of ICM in HNSCC was significantly increased (PD1 on CD4+ T cells: p = 0.001), while they found a decreased co-stimulatory molecule expression (e.g., CD27 on CD4+ T cells and CD39+ T 31 cells: p = 0.003 and p = 0.009, respectively). Furthermore, immune cells of HPV neg HNSCC have a significant age-dependent decrease of CD27 expression on CD8+, CD4+ and CD39+ T cells. It is a solid article with results that could potentially enrich current knowledge and clinical practice. I suggest few modifications. There are numerous limitations of this type of study and I encourage authors to explain in more details why these limitations could influence final result (i.e. authors focused mainly on identifying cytotoxic and immunosuppressive T cells, but did not include other subpopulations like NK-T cells - this could potentially influence final results). Also, what would be suggested direction for future studies in the light of your findings? Is there any clinical utility of these findings right now? Please cite literature with the similar topic as your study (compare results and emphasize similarities and differences).
Authors’ reply:
Given the limited number of markers that could be incorporated into our staining panel, we refrained from including an additional analysis of NK-T cells and instead focused on subsets of cytotoxic and immunosuppressive T cells. Nevertheless, it will be essential to examine immune checkpoint expression in the context of aging across additional immune cell subsets, which should be addressed in future studies. To provide further context, we have added a sentence at the end of the first discussion paragraph regarding age-related effects of immunotherapy in other solid tumors such as lung cancer and melanoma. We have also extended the limitations section of the discussion. In addition, future work could investigate changes in immune checkpoints during the course of radiochemotherapy, with particular attention to differences between age groups. To further elucidate the clinical relevance of our findings, it would also be important to assess patients’ responses to immune checkpoint blockade in relation to the presence of checkpoint molecules in peripheral blood prior to therapy initiation, as this may represent a prognostic factor before treatment. To address this, we have adapted the Conclusion section to clarify that our findings do not imply changes to the current treatment of elderly patients, as there are currently no age-specific treatment guidelines.
Round 2
Reviewer 1 Report
Comments and Suggestions for Authors
I apologize for not completing the re-review within the allotted time. The authors have adequately addressed my previous comments: where revisions were feasible, appropriate changes have been made; where changes were not practical, reasonable justifications were provided. I agree with the authors that, in clinical data collection, it is often difficult to obtain ideal, fully curated case datasets. In my view, the manuscript now meets the standards for publication.
Reviewer 2 Report
Comments and Suggestions for Authors
Authors have responded to all the queries.